# Environmental Impacts of Coal Nanoparticles from Rehabilitated Mine Areas in Colombia

**Marcos L. S. Oliveira [1,2,\*], Segun A. Akinyemi [3], Bemgba B. Nyakuma [4] and Guilherme L. Dotto [5]**

1   Department of Civil and Environmental Engineering, Universidad de la Costa, CUC,
    Barranquilla 080002, Atlántico, Colombia
2   Department of Sanitary and Environmental Engineering, Federal University of Santa Catarina,
    Florianópolis 88040-900, Santa Catarina, Brazil
3   Department of Geology, Faculty of Science, Ekiti State University, Ado Ekiti 362103, Nigeria;
    akinyemi70@gmail.com
4   Department of Chemistry, Faculty of Sciences, Benue State University, Makurdi 102119, Nigeria;
    bbnyax1@gmail.com
5   Department of Chemical Engineering, Federal University of Santa Maria,
    Santa Maria 97105-900, Rio Grande do Sul, Brazil; guilherme.dotto@cuc.de.co
*   Correspondence: marcos.oliviera@cuc.de

**Abstract:** With the possible increase in mining activities and recently projected population growth in Colombia, large quantities of nanoparticles (NPs) and potentially hazardous elements (PHEs) will be of major concern to mine workers, indigenous residents, and surrounding communities. This study highlights the need to regulate the pollution from Colombian mining activities that comply with regional regulations and global strategies. Colombian coal rejects (CRs) from the Cesar Basin, Colombia, were studied primarily by advanced electron microscopic and analytical procedures. Therefore, the goal of this research is to evaluate the role of NPs in the alteration of CRs' structure in a renewed zone at Cerrejón coal area (La Guajira, Colombia) through advanced electron microscopic (AEMs) methods. The objective of the analysis is to evaluate the incidence mode of nanoparticles, which contain potentially hazardous elements. The bulk crystallography (X-ray diffraction), chemical structure, and morphologies of NPs were studied by high-resolution transmission electron microscopy (HR-TEM), field emission scanning electron microscopy (FE-SEM), micro-beam diffraction (MBD), selected area electron diffraction (SAED), and energy-dispersive X-ray spectroscopy (EDX) procedures. The AEMs provided comprehensive insights into the geochemical evolution of CRs. Consequently, the AEMs can be used as essential tools for CR management in coal mining areas. The regular dimension of detected NPs was found to be above 2 nm. Ultrafine particles of quartz were identified by the advanced electron microscopy. Furthermore, the findings also revealed aluminium, calcium, potassium, titanium, iron oxides, and PHEs in the CRs. The extensive water practice in the coal extraction process combined with atmospheric oxygen supports oxidations of iron sulphide, thus releasing PHEs to the surrounding environment. Dehydration of sulphate salts fluctuate at consistent humidity in the coal mine environments. The study demonstrates the great influence of coal mining activities on the environment and human health.

**Keywords:** coal reject aggregates; Colombian mine pollution; Cesar-Ranchería Basin; nanoparticles; potentially hazardous elements

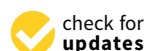



## 1. Introduction

Coal mining activities have resulted in multiple environmental and social impacts, such as waste management, deforestation, acid mine drainage, increase in noise levels, dust, contamination of local streams and wetlands, and contamination of soil profiles [1,2]. Global energy demand occasioned by continuous growth in "commodity frontiers" [3], along with novel styles of developing capital, and coal mining have been linked to potentially

hazardous elements (PHEs) and their disposal [4]. In addition, major mining corporations in the southern hemisphere have become global players [5], while industrial developments have recently further extended the developing frontiers of coal mining. As an effect, the previous years have seen a substantial intensification in socio-environmental conflicts connecting worldwide populations that combat coal mining activities [1]. The main coal-bearing Paleocene Cerrejón Formation is situated in the area of La Guajira, the Paleocene Los Cuervos Formation, found in the zones of Cesar and Norte de Santander and the upper Maastrichtian, to Paleocene Guaduas Formation, situated in the zones of Cundinamarca and Boyaca. Some authors have proposed that the studied area was formed in deltaic and intermediate environments [6,7] and had a noteworthy impact on the region [8]. The geology and petrographic features of the studied coal areas vary significantly, with coal ranks wide-ranging, from lignite to bituminous and anthracite [9].

In general, open pit coal mining includes removing the flora and shallow soil from a zone for coal exploitation; this mining activity contaminates the soil, rivers, drainages, and biodiversity [10–12]. Consequently, restoration agendas must guarantee the remediation of environmental resources in polluted areas [13,14]. The opencast coal mining technique usually takes place under life-threatening drought conditions in La Guajira (Colombian Caribbean region). After coal mining activities, a typical vegetation restoration has been applied as part of an effort to restore ecosystem functions in the abandoned mining areas [15]. The environmental recovery of Colombian coal mines is buried underneath spoil heaps protected by topsoil. This category of topsoil is classified as Technosol [16]. In the Cerrejón mine in Colombia, more than 70% of the topsoil (>100 cm) is instituted by a spoiled coal mine. The current research from Cesar-Ranchería Basin (CRB) (Figure 1) consists of coal from the Mid-Late Paleocene Los Cuervos and Cerrejón Formations. Coal mine rejects are aggregate waste resulting from washing coal, with substances such as coal fines, soil sand, and rock resulting from the process. The coal mine rejects contain a lower percentage of coal. Coal mine reject aggregate is usually used as a replacement of natural aggregate in granular sub-base (GSB) and bituminous layers in road construction [17].

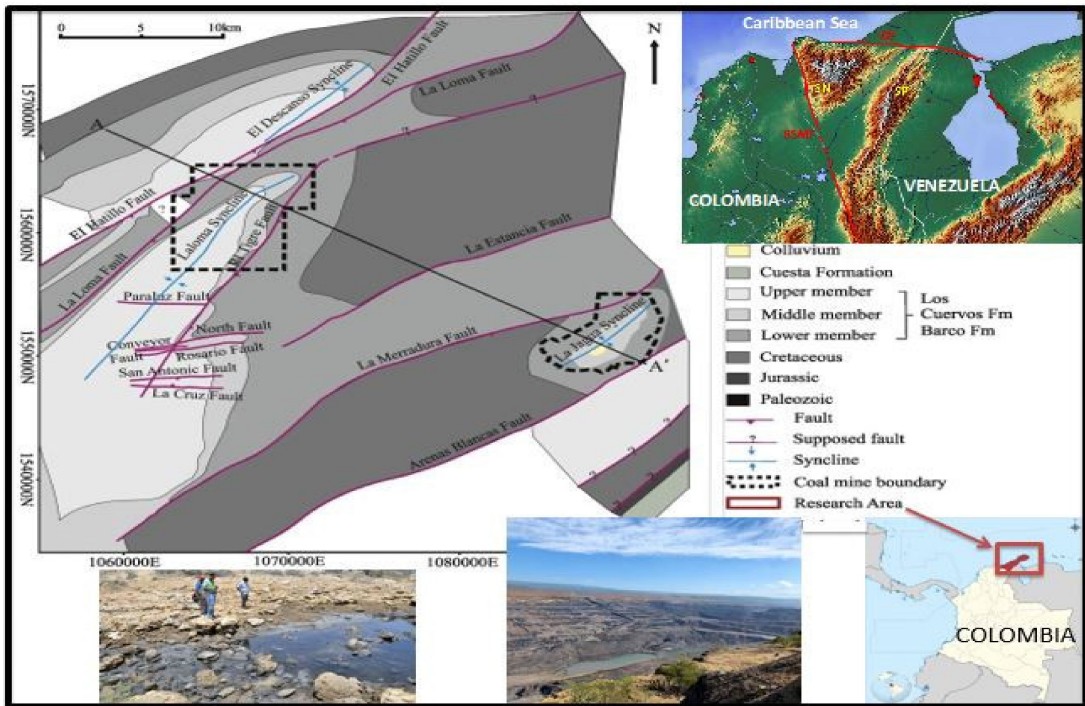

**Figure 1.** The area of the coal basin studied.

Poor disposal of coal mine rejects (coal waste) signifies substantial environmental anxieties due to their possible impact on neighbouring coal industrial areas. Coal mine

rejects from active and/or abandoned mines are sinks for high concentrations of potential hazardous elements (PHEs) and nanoparticles (NPs, minerals and/or amorphous compounds) [18]. The unlimited concentrations of PHEs in Colombian (La Guajira area) sources of coal mine rejects would result in yearslong leaching of PHEs into rivers and drainages. Rabha and Saikia [19] reported that there is a positive correlation between long time inhalation of coal-derived nanoparticles and increased health risk. Some authors have reported the occurrence of nanominerals, nanoparticles and potential hazards in feed coals and combustion residues [20–22]. Nevertheless, the chemical nature and compositional information about the nanominerals and nanoparticles inhaled in coal mine rejects hitherto remain scanty [23,24]. The aim of this study is to demonstrate in a scientific way how harmful coal extraction could be for the environment and native communities. Therefore, this study is designed to assess the complexes in NPs and UFPs of the Guajira coal region in order to determine the environmental and human health risk. The main objectives of the research are to (1) apply advanced electron microscopy for identifying coal reject (CR) aggregate types acquired from Cerrejón coal zones with varying years of recuperation, and (2) understand the influence of NPs on the quantity of PHEs in the soil, and speciation movement in CRs. This is an innovative study conferring the properties of NPs on the geomobility of PHEs in CRs. The acquired data provides an empirical basis for highlighting the environmental hazards caused by coal mining on topsoils in Colombia.

## 2. Study Area

The total area granted for the exploration and exploitation of mineral resources, such as coal mining areas in Colombia, is about 4 million ha [25]. In 2017, coal mining GDP reached around USD 549 million, or 65% that of mining GDP, which represents about 1.24% of the total GDP of Colombia. The total value of coal exports in 2017 was USD 1.55 billion [26]. As the fourth largest coal exporter in the world, Colombia is susceptible to global market trends, since over 90% of production is exported [27], although some products are used locally. In the future, the Colombian government projects significant growth in coal production, around 105 Mt by 2024, with plans to export 97 Mt (i.e., 93% of net extraction) [28].

Despite the widespread environmental and social impacts of the mining industry, coal market analysts place Colombia in a strong competitive position, owing to low production costs and high-quality coal [29]. However, market trends and improved climate policies do not suggest promising prospects for future exports. Therefore, the government must prepare for these contingencies since the economy of Colombia is highly dependent on coal exports. Furthermore, increased competition in the Atlantic and Pacific markets will keep coal prices low, which could force mining companies operating in Colombia to abandon operational guidelines that safeguard human health and environmental protection. Consequently, Colombian authorities must prioritise environmental conservation against the backdrop of increased mining production, new economic realities, and local externalities [2].

The Cerrejón coal mine originally began operations in the 1970s as a cooperative between Exxon Corporation and the publicly owned Carbocol [2]. However, the coal mine area was wholly denationalised as part of Colombia's neoliberal reorganisation in the year 2000. As such, the Colombian public's 50% stake in the Cerrejón was sold to three transnational companies: Anglo American (UK/South Africa), BHP Billiton (UK/Australia) and Glencore Xstrata (Switzerland). Colombia is currently the fourth largest exporter of coal in the world, with 90% in exports [30]. Similarly, about 60% of all coal mined from Cerrejón is exported to Denmark, France, United Kingdom, Germany, the Netherlands, Portugal, Spain, Ireland and Turkey [2].

The coal study area (Figure 1) is located in the La Guajira Department (11°3′ N, 72°44′ W–11°8′ N, 72°37′ W). The Cerrejón and Perijá thrust faults consist of the eastern and the Sierra Nevada de Santa Marta massif in the north-western border [31,32]. The area consists of Precambrian to Palaeozoic igneous and metamorphic rocks [33]. The

geochemical record indicates that it is made up of approximately 2 km thick clastic basin-fill varieties from the Palaeozoic to Holocene era [31,33]. The surrounding environment is made up of cyclical, dry, and tropical vegetation. The weather is dry with an average annual temperature of 27 °C. Rains occur between March–April and October–November with a yearly mean of 800 mm and evapotranspiration from 950 to 1550 mm. The soil is categorised as Entisol, which shows more aggregates of kaolinite and ferric and/or aluminium oxide at deeper layers [34].

*Social Context*

Coal mining has a legacy of environmental injustice in some countries, particularly in Colombia, where the exploitation of coal resources has not been conducted in a sustainable manner. In the country, the mining industry is known for corruption, environmental impacts, and social conflicts [35]. Correspondingly, Pérez Rincón [36] reported that coal mining represents about 15% of the environmental conflicts in Colombia. Likewise, Cardoso [37] stated that every ton of mined coal produces socio-environmental liabilities at each stage in the life cycle of the process. Moreover, along with the coal chain, environmental merits and demerits are accrued and, ultimately, generate ecological-distributive conflicts [38]. The coal-mining industry has also displaced multitudes of ethnic, country-dwelling, and Afro-Colombian populations in the area [37]. The official dislodgment of these groups is based on the legal procedure called "expropriation" in Colombia. Therefore, the government supports or partakes in the displacement of people in regions selected for megaprojects or the extraction of Colombia's natural resources [2]. However, the extractive model of Colombia coexists with the state's commitment to the 1991 Constitution, which grants the right of prior consultation to indigenous people [39]. In addition, Colombia endorsed the International Labour Organization (ILO) Convention No. 169 on the rights of indigenous peoples [40]. Despite these seemingly progressive laws, the rights of these communities, as reported by other authors [41], are continually subordinated to the "rights" of multinationals to make a profit. Therefore, rights granted to local and traditional communities and policies to protect ecosystems and biodiversity are affected by licences for mining exploration [42,43]. Consequently, coal mining in La Guajira has produced severe injustices. Healy et al. [2] summarised several injustices denounced in communities affected by mining operations and highlight that the inhabitants of mining regions have become victims, instead of recipients, of the mining of resources. Furthermore, communities are expatriated continuously by coercion, physical force, and intimidation, or through the pollution of farmlands and drinking water, and malnutrition. Moreover, the availability of drinking water is a crucial concern in the areas periodically affected by drought. The rate of water consumption in Cerrejón is about 17 million litres daily, whereas access per person, on average, is about 0.7 litres daily. Currently, about 30,000 persons work directly for the three largest coal corporations in Colombia [44]. The estimates of the coal reserves in Colombia differ considerably in the literature. The coal reserves are projected to be between 5 and 6.4 Gt [45], which is less than 2% of worldwide reserves. The remaining carbon reserves cannot drive the speedy removal of coal, since it is an abundant resource in Colombia and worldwide. Since most carbon resources are predicted to remain underground, the redundant investment must be readdressed to avoid irreversible loss of assets [27].

## 3. Materials and Methods

### 3.1. Sampling Strategy

Coal reject samples from different disposal dumps with various ages in the La Guajira Department were collected from the topsoil. The CRs were obtained and directly divided by a Teflon circle in 1 cm intervals until a depth of 20 cm, followed by 2 cm intermissions up to a 40 cm distance. This was to prevent physico-chemical alterations caused by direct contact with atmospheric conditions. All CRs were stored in a cold, secluded environment to avoid post-sampling geochemical alteration, and stored at room temperature before laboratory analysis. Five zones, recovered from 3-, 9-, 15- and 21-year old coal reject dumps, were

chosen after chronosequence (similar procedure previously reported by Silva et al. [46]), and one forested zone unaltered by coal mining was sampled as a "blank" zone. This strategy is typical in works directed in rehabilitated coal zones. Twenty-five CRs from coal seams were sampled from coal mines (Figure 1). Nineteen CRs were sampled from six coal seams inside the Cuervos Formation, and the other CR samples were taken from different coal seams. All CRs were sampled roughly at each meter. Quantities were separated by chosen portions and CRs powdered (<200 μm), correspondingly, for AEM analyses.

### 3.2. Analytical Methods

#### 3.2.1. Mineralogical Phase Analysis

The phase analysis of CRs was conducted using X-ray powder diffraction (XRD, PW1830, Phillips) diffractometer with Cu Kα radiation. Before each run, the powder was smoothened using a slide to obtain a uniform thickness suitable for the X-ray beam, before scanning from 2 to 60° for 2 h.

#### 3.2.2. Morphological Analysis

The Raman spectroscopy (RS) was operated in the Renishaw model Invia Reflex Raman scheme based on the confocal mode [14]. The principal Raman Spectroscopy tests were performed on polished epoxy-bound pellets applied for AEMs investigation. The spectra acquired from CRs were used to analyse the Raman factors and regulate the precise incidences as well as the relation intensities of the bands of the C particles.

#### 3.2.3. Electron Beam Microscopy

Field emission scanning electron microscopy (FE-SEM) fitted with a turbo pumped chamber, a motorised stage, an Oxford energy-dispersive X-ray (EDS) spectrometer with a resolution of N133eV and a four-quadrant back-scatter detector, was used to identify the minerals by conventional SEM techniques. This was based on observation of whole specimens using natural and polished surfaces, and a high-resolution transmission electron microscope (HR-TEM, 200 kV) equipped with an efficient FE cathode. The setup also contained energy omega-filter selected area electron diffraction (SAED), and Fast Fourier transforms (FFT). Lastly, tests were also performed using scanning transmission electron microscopy (STEM), micro-beam diffraction (MBD), and energy-dispersive X-ray spectroscopy (EDS) techniques [47–49].

High atomic number and low atomic number elements were observed in the brightest regions and the dark-field zones of the TEM image [21]. Therefore, the presence of PHEs and NPs in CR samples was confirmed.

#### 3.2.4. Quantitative Mineral Phase Analysis

The Siroquant analysis technique was used to estimate the percentage of non-crystalline (amorphous) material present in the ash samples, based on comparative analysis with a poorly crystalline silica phase (similar to, but not equivalent to, tridymite). The exact sizes used for analyses have been reported earlier in the literature [10,46,47]. The samples were examined through the dividing visual microscope of Nikon® SMZ645®. The analytical grain mounts for examination under the light microscope were prepared with Cargille® Meltmount®. Next, the slides were examined with a Nikon® Labophot2-Pol® optical microscope at 10×, 20×, and 40× objectives. The Nikon® Digital Sight DS-SM® camera and interface were subsequently employed to obtain the photomicrographs of each phase. The refractive indicators were examined by Cargille® Certified Refractive Index Liquids Series A and B for the non-fibrous phases. Furthermore, the significant solid-phase regions were examined for marginal bleaching with the Spectral Confocal and Multiphoton Microscope Leica Model TCS SP2 [32].

## 4. Results

For the nine-year detected CR aggregates, the area shows a grade of restructuring within the topsoil assembly. Composed with particular planar voids, a different porosity was formed due to a marked intensification of the in situ organic occupation. The CRs 21-year zone presented a new progressive stage of topsoil relationship, in comparison to the preceding 9-year-old topsoil. The study found CR aggregates integrated with quartz (Figure 2), which indicates that a strong aggregation was detected in several zones.

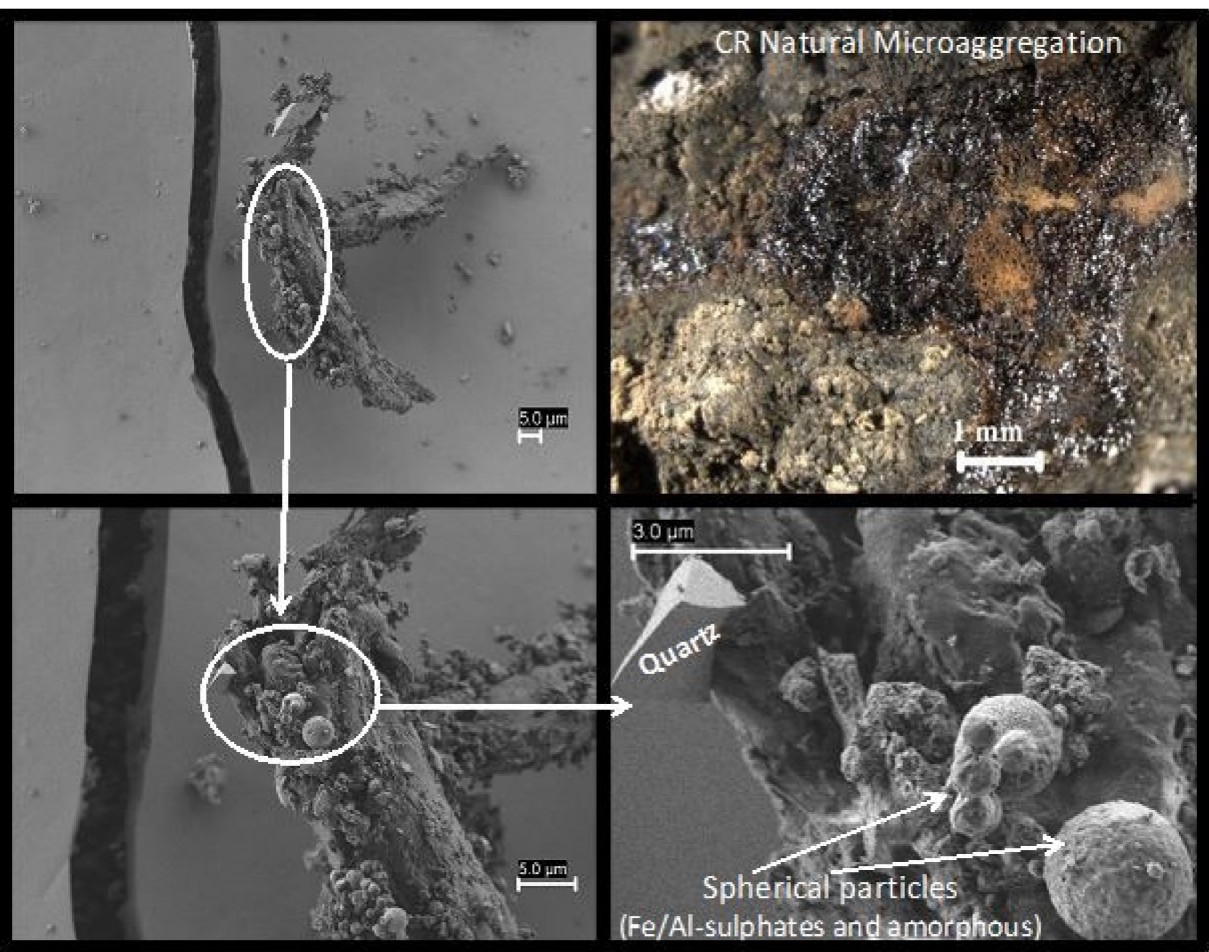

**Figure 2.** Selected CR aggregations with quartz, sulphates and amorphous phases.

For the topsoils examined, there was a visual development arising from sulphide oxidations, carbonate and clay dissolution, sulphate and oxide generation, regular loose infilling, and modification of soil elements. Thus, primary phases, due to the creation of combined aggregates and improved pore organisation in the 21-year area, were detected. The transformations detected in the structure and the textural CRs are associated with biotic action and the minerals' geochemical reactions.

*Major Minerals and NPs*

The mineralogy of the Colombian CRs varies moderately. With some notable exceptions, the phases identified in the CRs are typically found in CRs from several countries [29,46,48]. Evaluations of XRD dates and the main phases in CRs are: clays, sulphates, oxides and sulphides. These phases are principally composed of quartz, kaolinite, illite, smectite, jarosite, gypsum, hematite, rutile, pyrite and other comparable phase assemblages previously reported in other coal studies [11,14]. Numerous minor accessory species also detected by AEMs include: anhydrite, arsenopyrite, barite, brucite, calcite, dolomite,

epsomite, galena, gibbsite, goethite, hexahydrite, marcasite, melanterite, monazite, natro-jarosite, siderite, sphalerite, rosenite, rutile, schwertmannite, and zircon (as in Figure 3). Quartz was the most abundant mineral detected by XRD. Consequently, the ultrafine particles (UFPs) of quartz were easily observed by HR-TEM/SAED/EDS (as in Figure 4). Ultrafine particles of quartz, depending on the size and shape, can be easily aerosolized and inhaled.

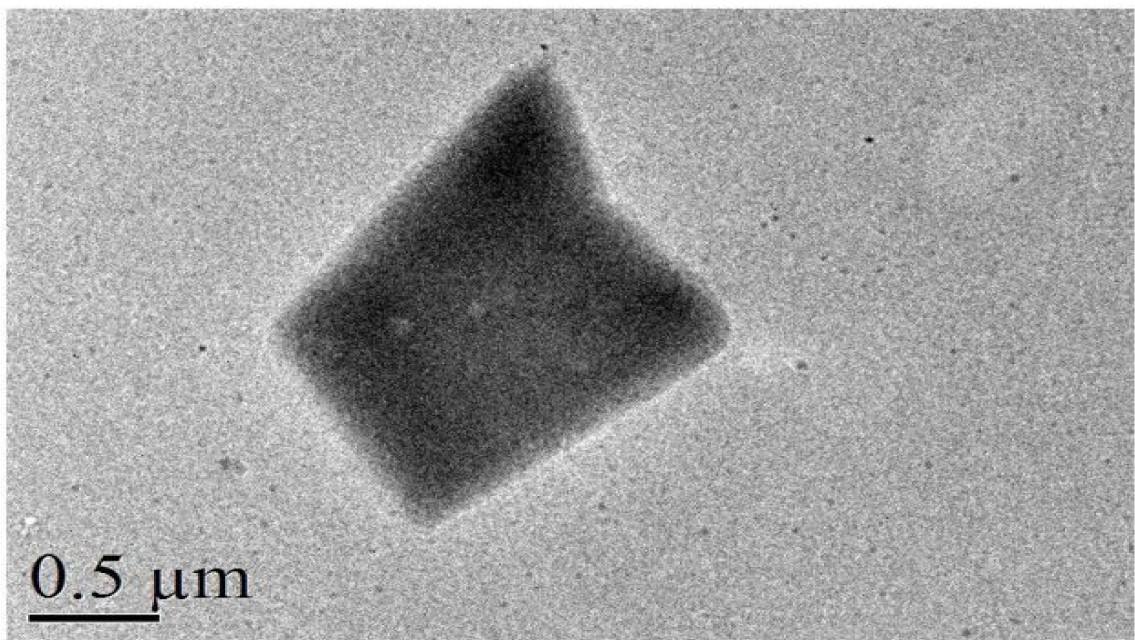

**Figure 3.** FE-SEM image of zircon (and Zr, Si, and Al mapping).

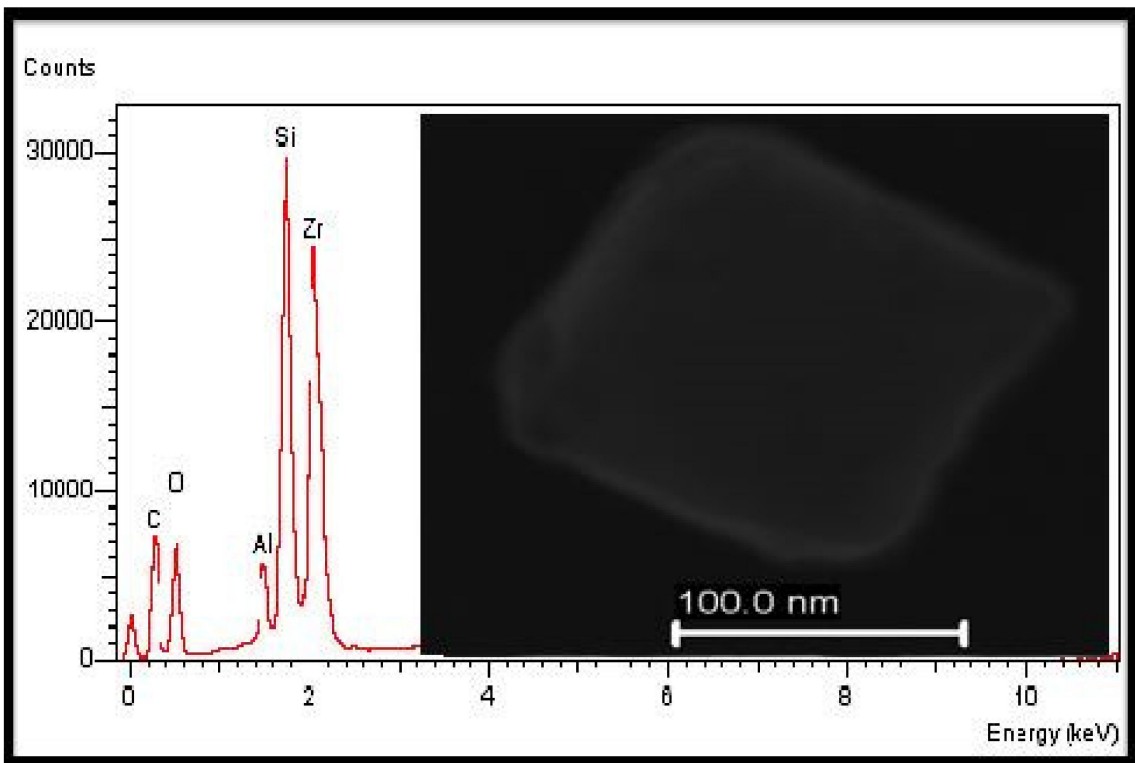

**Figure 4.** Typical ultrafine quartz detected in Colombian CRs.

Moreover, numerous research studies have been motivated by the surface geochemistry of phase assemblages isolated from mining environments [46,47]. Primary minerals produced directly from Colombian mining practice are typically sub-micrometre (0.1 nm to 10 μm) of solid phases, with PHEs (Figure 5).

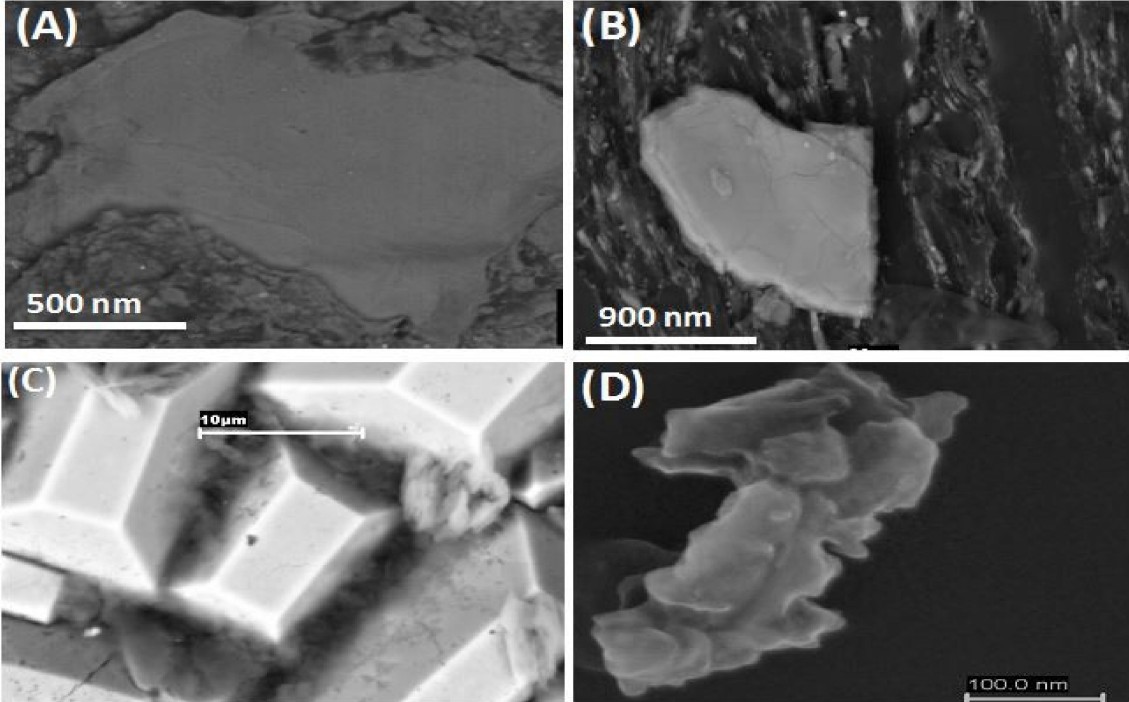

**Figure 5.** Classical ultrafine primary minerals. (**A**) Clorite; (**B**) Calcite; (**C**) Pyrite; (**D**) Siderite.

The PHEs within separate CRs are not regular (e.g., Zr in zircon, from Figure 3). In particular, mixed elemental allocations of aluminium, calcium, iron, oxygen, potassium, and silicon in iron silicates were detected by investigative ultrathin cross-partitioned CRs. Such iron minerals (microscopic, ultrafine, and NPs) display very crystalline characters (e.g., hematite, Figure 6A), as demonstrated by HR-TEM/EDS. Hence, the Fe particles were examined by electron diffraction patterns of the crystalline compounds through SAED and MBD techniques.

The detected NPs containing PHEs (e.g., magnetite and jarosite, Figure 6B,C) were used as reliable intermediates for chemical recording in this study. This is because these species offer a compound sample of the catchment Colombian coal zone, upstream of the sampling area. This combination of rock and topsoil performance is a decisive sink for PHEs resulting from within the catchment. Likewise, it reflects the effects of PHEs on alterations of ecological circumstances, which could pose contamination complications [49].

The intensive water usage in Colombian coal extraction, combined with atmospheric oxygen exposure, benefits sulphide reactions. The sulphides of iron react with water and $O_2$ to generate sulfate and Fe-oxyhydroxides [24]. The detected iron sulphide oxidations may increase at the location of the PHEs, typically introducing: arsenic, mercury, selenium, or lead, among others (e.g., Figure 6). Similar results were reported by Civeira et al. [10]. Numerous phases containing F, Cr, Cd, Sb, U, and other PHEs were also detected in the CRs. These phases vary with the CRs. The NPs of Fe-sulphate and alunite are mostly detected in coal zones in Colombia. Many of the aggregates examined (as in Figure 2) and detected were monomineralic and covered by gypsum, iron sulphates and oxides, beyond abundant amorphous compounds.

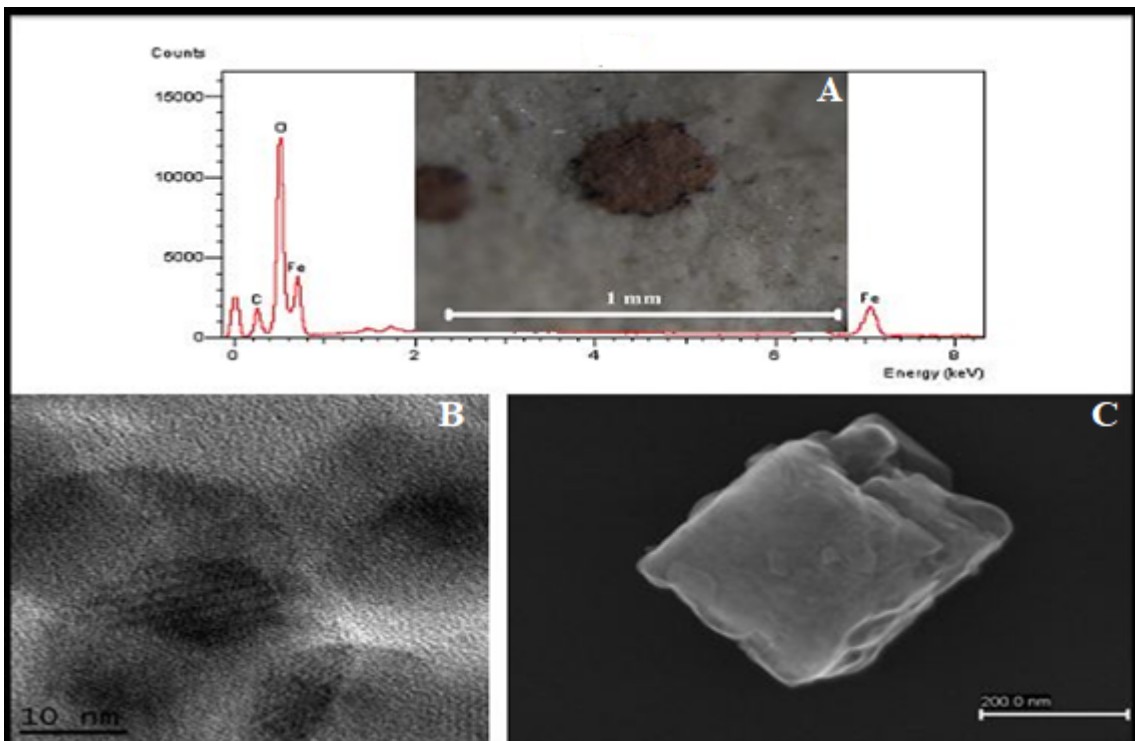

**Figure 6.** Selected iron minerals. (**A**) Hematite and EDS; (**B**) magnetite containing As and Se; (**C**) jarosite containing Pb.

The CRs analysed by HR-TEM/EDS and FESEM demonstrated the prevalence of $K^+$ over $Na^+$ in jarosite, although natrojarosite was also detected. Several detected Fe-sulphates particles (e.g., jarosite) can assume a cubic form, signifying that it is generated as a pseudomorph after Fe-sulphides. Many Mg-Mn-Fe-Al salts (e.g., halotrichite, hexahydrite, epsomite, and pickeringite) are also easily detected in the studied Guajira coal regions. The dehydration of salts differs based on the humidity. The FE-SEM/EDS data obtained validates the moderately complicated behaviour of these salts, particularly in the 21-year areas. In this circumstance, particular reflection is needed as the vacuum AEMs can produce dehydration of salts from events in the coal area.

Furthermore, acidic phases with Fe-sulfates were produced during CRs' exposure to the atmosphere. Ultimately, this can contaminate water, soil, and sediments from the studied Colombian coal area. The selected nanophases studied by AEMs were created to understand the PHEs, such as Se, Hg, Pb, and As soluble phases (e.g., salts). Comparable data have been studied by other authors in the literature [50]. Regarding the extraordinary geomobility, NPs containing PHEs are measured due to their ability (low boiling points) to spread contaminants worldwide [10,14,29] by alternating between geochemical systems and contribution to biotic cycles. In addition, particular new works showed that mercury could also be retained in the organic compounds, creating a double combination, transferring at a greater scale and affecting the swimming mobility in a different manner [51].

Copper, chromium, vanadium, and nickel can substitute iron in minerals and amorphous phases in the CRs of the studied area. The occurrence of arsenic is mostly by arsenopyrite distribution in the Colombian CRs rocks. Arsenopyrite consumption is related to the discrete sulphide mineralisation occurrence, which is different from the chief chalcopyrite sulfidation in the analysis zone. The destructive influence of pH in this issue suggests that an acidic ecosystem is essential for the liberation of the PHEs from their environmental constituents. The Sc is probably linked with organic matter from CRs. Its small magnitude and great control help the generation of constant organic compounds in the studied topsoils and reactions in clays. The metalliferous constituents of the detected NPs are probably geochemically multifaceted. The AEMs data in the present study proves

that at least some amorphous NPs, such as the iron clay particle indicated in Figure 6, have a varied structure in which aluminium, iron, PHEs, and silicon are not uniformly allocated.

## 5. Discussion

Modifications in meiofaunal abundance, variety, and circulation are among the widely explained effects of human alteration, while precise answers and their course may be essentially alterable [52,53]. Preliminary theories were moderately reinforced in terms of multivariate alterations, which are linked with distance to the channels [54]. Residents in coal mine zones are frequently exposed to hazardous multifaceted combinations of coal compounds.

The chemical structure of potentially hazardous elements (PHEs) is demonstrative of the drainage, topsoil, and waters around the coal basin [11]. The environmental weathering of studied phase deposits, as well as coal mining, has consequences in the high proportions of PHEs in CRs. The exhaustive water usage in the Colombian coal extraction process, coupled with the chemical interactions of atmospheric oxygen, supports sulphide reactions. The iron sulphide oxidations (acidic phases) promote the release of PHEs, typically arsenic, mercury, selenium or lead, among others, into the coal mine areas. The unlocked potentially hazardous elements (PHEs), due to iron sulphide oxidations, can pollute the water, soil, and sediments from the studied Colombian coal area. Some detected Fe-sulphate particles (e.g., jarosite) are generated as a pseudomorph after Fe-sulphides. Dehydration of numerous sulphate salts is dependent on regular humidity. Generally, in times of higher humidity and rainfall, hydrated minerals are detected, while in drought, and in the dry season, the minerals are less hydrated or dehydrated.

Considering the numerous applications of quartz, it is crucial to examine the impact of ultrafine particles (UFP) of quartz on human health, safety, and the environment [55]. Contact with microscaled particles of quartz has been related to the prevalence of numerous medical conditions, such as lung cancer. It is also widely recognised that microscale quartz, when aerosolized, causes severe health hazards, such as silicosis from micron ranging fibres [56]. However, numerous cytotoxicity studies performed on silica nanoparticles revealed prospects of severe health hazards by encouraging inflammation and oxidative stress. Hence, the application of nanoparticles in industrial surroundings could result in acute exposure to humans [57]. The inhalation of NPs of CRs is reportedly the principal path of exposure to PHEs in working [9,58] and ecologically exposed people. Consequently, the principal obstacle to the inhalation of CR particles is buccal cells [59]. These cells have been examined as a suitable objective position for observing people who have been in contact with occupational and ecological genotoxins. Most geochemical composites originate in hazardous combinations created in exposed cast CRs and enter the body through absorption [8]. Therefore, epithelial cells positioned in the higher aero-digestive region are the principal obstacle for inhalation or ingestion, since the cells are extremely proliferative [60,61]. These are comparable to reactive oxygen species, which qualify as necessary for cellular macromolecules and DNA [39].

The nanophases of amorphous minerals are naturally formed by environmental forces [22,23,62,63]. Therefore, toxicity could be intensified by a decrease in particle size, from micro to nanoscale. This has improved the foundation of amplified superficial total atoms, as reported by Kumar et al. [63]. In the present study, high-spatial-resolution methodology, using AEMs as the principal investigations for both chemistry and assembly, were applied to examine ultrafine particles and NPs containing PHEs. The AEMs data on the nanophases and UFP from Colombian CRs, which are not obtainable from XRD research, have consequences on human health and the environment.

## 6. Conclusions

This study presents a general assessment of the complexes in NPs and UFPs of the Guajira coal region, which can be associated with the mining areas from other countries.

It similarly proposes an alignment dataset for extra works of associations between the ecological issues and social impacts in numerous coal regions worldwide.

The XRD, Raman, and AEMs data confirmed the existence of amorphous, crystalline, and carbonaceous nanophases with PHEs (e.g., arsenic, cadmium, chromium, mercury, lead, and other elements). The intense water usage in the extraction process with atmospheric oxygen promotes oxidations of iron sulphide, unlocking PHEs in the surrounding environments. The AEMs analysis recognised some unidentified amorphous NPs in coal zones in Colombia. The ultrafine particles of quartz, as detected by advanced electron microscopy, are ascribed to the prevalence of numerous medical conditions, such as lung cancer, in the coal mining environments. Dehydration of various sulphate salts in rehabilitated coal mine fluctuate, depending on humidity. The results of the study emphasise that CRs contain nanominerals that host potentially hazardous elements that are harmful to the environment and human health. Further studies should involve the consideration of toxicological risks in ecological CRs from coal mine zones for contrast and the calculation of danger to ecological health.

**Author Contributions:** M.L.S.O.: writing—original draft: Data Curation. S.A.A. and B.B.N.: writing—review & editing. G.L.D.: Resources. All authors have read and agreed to the published version of the manuscript.

**Funding:** This research received no external funding.

**Institutional Review Board Statement:** The study was conducted in accordance with the Declaration of Helsinki, and approved by the Institutional Review.

**Informed Consent Statement:** Informed consent was obtained from all subjects involved in the study.

**Data Availability Statement:** Not applicable.

**Acknowledgments:** This research work and data analysis were carried out with the assistance of the CUC and Conselho Nacional para o Desenvolvimento Científico eTecnológico (CNPq). The authors also acknowledge the Colombian coal communities for access to the samples of the CRs.

**Conflicts of Interest:** The authors declare that they have no known competing financial interests or personal relationships that could have appeared to influence the work reported in this paper.

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
