# Peer review of "Environmental Impacts of Coal Nanoparticles from Rehabilitated Mine Areas in Colombia"

_sustainability, doi:10.3390/su14084544_

Round 1

Reviewer 1 Report

The Authors submitted a paper on the “Environmental and social impacts of coal nanoparticles from rehabilitated mine areas in Colombia”. The subject of the paper is fully coherent with the topics covered by Sustainability and, specifically, with those of the Thematic Issue to which the paper was proposed. However, several orders of problems emerged during the review:

1) The introduction of the study area has to be consistently improved, both for the geology and for description of mining and rehabilitation activities in the sites – both aspects are essential for the development of the discussion and to find the link with the environmental and social consequences of mining;

2) Sampling and sampling methods are not sufficiently explained, I think that a sampling map and some images may be helpful;

3) the Authors display an impressive set of analytical methods and devices, but where (in which laboratories) the analyses have been performed?

4) most important of all, Results are not clearly separated from Discussion, and appropriately presented, generating confusion in the reader; more figures are required, and figures and captions proposed by the Authors are of limited significance;  moreover, despite the large set of analytical devices, data presented are mostly qualitative, so that their support to discussion is partial and questionable;

5) consequently, conclusions do not appear supported by data and by an effective discussion;

6) one more relevant point, English must be carefully revised, as many sentences are not clear or badly expressed.

For all those reasons, I think that the paper is at this stage not sufficiently mature for publication in Sustainability:

I suggest the Authors to improve appropriately chapters 1,2 and 3 and to totally rewrite and expand the text in chapter 4, clearly separating the results from discussion, and integrating the results with quantitative data; obviously, this also requires to rewrite the conclusions . Further notes and suggestions may be found in the attached text.

Author Response

Response to Reviewer’s #1

1) The introduction of the study area has to be consistently improved, both for the geology and for description of mining and rehabilitation activities in the sites – both aspects are essential for the development of the discussion and to find the link with the environmental and social consequences of mining;

Author’s Response:  Information on the geology, description of mining and rehabilitation activities has been provided under the introduction section of the manuscript.

2) Sampling and sampling methods are not sufficiently explained; I think that a sampling map and some images may be helpful;

Author’s Response: Map of the study area is shown in Figure1 and sampling methods is succinctly explained in section 3.1.

3) The Authors display an impressive set of analytical methods and devices, but where (in which laboratories) the analyses have been performed?

Author’s Response:

4) most important of all, Results are not clearly separated from Discussion, and appropriately presented, generating confusion in the reader; more figures are required, and figures and captions proposed by the Authors are of limited significance;  moreover, despite the large set of analytical devices, data presented are mostly qualitative, so that their support to discussion is partial and questionable;

Author’s Response: Results are separated from the discussion. Figures and captions are well cited with the text. Discussion has been significantly improved upon.

5) Consequently, conclusions do not appear supported by data and by an effective discussion;

Author’s Response: The conclusions drawn from the study and discussion are supported by presented data.

6) One more relevant point, English must be carefully revised, as many sentences are not clear or badly expressed.

Author’s Response: English has been substantially edited by an English native.

For all those reasons, I think that the paper is at this stage not sufficiently mature for publication in Sustainability:

I suggest the Authors to improve appropriately chapters 1, 2 and 3 and to totally rewrite and expand the text in chapter 4, clearly separating the results from discussion, and integrating the results with quantitative data; obviously, this also requires to rewrite the conclusions. Further notes and suggestions may be found in the attached text.

Author’s Response: The authors have improved on the conclusion section of the manuscript.

Reviewer 2 Report

The article is correct from the point of view of the topic and importance of the problem. The situation with mismanaged mining in Colombia as well as in other Latin American countries is undoubtedly serious.
Nevertheless, the authors did not avoid mistakes regarding subjective opinions about the global harmfulness of coal mining. These parts of the publication (marked in the attached manuscript) should be edited.

Author Response

The authors have revise the manuscript based on the comments in the attached marked pdf copy.

Reviewer 3 Report

1. It seems to me that the list of abbreviations should be given as a separate paragraph, because the abstract is obviously cluttered with them.
2. In the "conclusions" section, only the first three sentences are of any scientific value. Everything else is superfluous.
3. In general, I do not have a clear understanding of the "integrity" of the article. The statement of the problem is unclear, although the relevance is undoubtedly there. But the purpose of the work is not indicated at all.  
4. The methods are described fully enough. The results of the analysis are also presented quite fully. However, there is no understanding of how these results can be used further, in other words - for what purpose this rather large work was done?
5. There is no consistency with the title - what social impact are we talking about? There is nothing in the article about that. About environmental - there is. It seems to me that the title of the article could be changed.
In general, I believe that the article can and should be improved and then accepted for publication; therefore, my current grade is "Reconsider after major revisions".

I wish you fruitful work on improving the article.

Author Response

Reviewer’s #3

Comments and Suggestions for Authors

  1. It seems to me that the list of abbreviations should be given as a separate paragraph, because the abstract is obviously cluttered with them.

Author’s Response: The authors have reduced the number of abbreviations in the abstract section.
2. In the "conclusions" section, only the first three sentences are of any scientific value. Everything else is superfluous.

Author’s Response: The conclusion section has been improved upon.
3. In general, I do not have a clear understanding of the "integrity" of the article. The statement of the problem is unclear, although the relevance is undoubtedly there. But the purpose of the work is not indicated at all.  

Author’s Response: The purpose of the study is provided under the introduction section.
4. The methods are described fully enough. The results of the analysis are also presented quite fully. However, there is no understanding of how these results can be used further, in other words - for what purpose this rather large work was done?

Author’s Response: The purpose of the study has been provided in the introduction section and conclusion section.
5. There is no consistency with the title - what social impact are we talking about? There is nothing in the article about that. About environmental - there is. It seems to me that the title of the article could be changed.

Author’s Response: We have changed the title of the article.
In general, I believe that the article can and should be improved and then accepted for publication; therefore, my current grade is "Reconsider after major revisions".

Reviewer 4 Report

OVERALL ASSESSMENT

The paper addresses an important topic that is currently very relevant and worth publication. However, some aspects need to be addressed/complemented.

In terms of the English language, some parts of the text are hard to follow and need editing.

Overall, I think that the paper is lacking more details and data analysis in the results and discussion section. The paper offers a very detailed qualitative description of the composition, characteristics, end effects of CRs, but falls short providing quantitative results, which are key to drawing conclusions and explaining the social-environmental impacts objectively.

In the following sections, I will provide a more detailed explanation of the opportunities for improvement.

ABSTRACT

The structure of the abstract can be improved. Instead of starting with the methods in the first two sentences, start with the background/problem statement (third sentence). Additional background information will make the abstract stronger. This is an example of a better structure for your abstract:

With the possible increase in mining activities and recently projected population growth in Colombia, large quantities of nanoparticles (NPs) and potentially hazardous elements (PHEs) will be of major concern (why is it a major concern? You can be a little more specific here) to indigenous residents (only indigenous residents? What about nearby communities?)...

After this opening, I recommend adding a sentence that includes a little more background (e.g., importance/benefits of gaining knowledge of CRs characteristics or constituents).

...This study highlights the need to regulate the pollution from Colombian mining activities that comply with regional regulations and global strategies. Colombian coals rejects (CRs) from the Cesar Basin Colombia, were studied primarily by advanced electron microscopic and analytical procedures. Therefore, the goal of this research is to evaluate the role of NPs in the alteration of CRs structure in a renewed zone at Cerrejon coal area (La Guajira, Colombia) through advanced electron microscopic (AEMs) methods. The objective of the analysis is to evaluate the incidence mode of nanoparticles (NPs), which contain potentially hazardous elements (PHEs)...

From here the structure of the abstract looks good.

Note: Remember to always write out all the first in-text references to an acronym, followed by the acronym itself enclosed by parentheses. 

1. INTRODUCTION

Line 33. I believe the first sentence is a serious statement that can be misinterpreted and considered subjective. To avoid that, I suggest using facts and data to back up this point. Instead of saying "Coal mining is one of the worst human and environmental tragedies on the planet" you can say something like: "Coal mining activities have caused several environmental and social impacts such as..." and highlight some examples of disasters, or tragedies that unfortunately are not rare in this industry.

Lines 45-57. What is the goal of these sentences? Why are you giving me this information? How does it relate to the understanding of CRs? You might want to add a sentence to elaborate on this.

I wanted to see what has been done in this field. Has anyone tried to study the coal rejects to identify potential hazardous elements before? Are the results from those studies consistent with your results? If there are no such studies, I think this is a great opportunity to elaborate on the importance of the study and your contributions to the field. I think that you kind of address this in line 53 but you should objectively say that no work has been done in this area.

Good job stating the objectives of the study very clearly at the end of the introduction.

Line 51. Here you introduce CR aggregates but what is their importance? Why are you studying the CR aggregates? Do they tell any valuable information about CRs? In the following sections of the paper you address the aggregates again but it is is unclear why you are studying them. Elaborate more on this. 

2. STUDY AREA

Line 97. Did you mean soil or earth? "The earth is categorised as Entisol, which showed..."

3. MATERIALS AND METHODS

3.1 Sampling Strategy

It would be great to have a figure here. It will be helpful to fully understand how the samples were collected and what methods were used to that end. I recommend using a flow chart ideally with images or illustrations showing the sampling procedure step by step. That will make the methods more reproducible as well. How does the Teflon circle look like? How big it is? It is hard to understand the 1 cm intervals, depths, and 2 cm intermissions without a graphic that explicitly shows it. If no figures/graphics are available, elaborate more to make sure your procedures are very clear and easy to follow.

Line 146. What do you mean by 3, 9, 15, and 21 years? How did you determine these years? Is this the time that the CRs have been sitting in that specific zones? You want to be very clear so there is no room for misinterpretation.

Line 149-151. The CRs from coal seams are not clear for me. Did you take material directly from the coal seams at the mine site? Did you take samples from material that was rejected but not disposed yet? What criterion did you follow to classify the material from the coal seam as CRs? Elaborate more to clear any possibility of doubt here.

Line 152. The word "showed" seems odd here. "Quantities were showed separated? on elected portions and powdered..."

3.2 Analytical Methods

You used an incredibly large number of methods here which is great, but it looks very confusing and disorganized. Why are you applying these different methods in the first place? Do you want to compare the results of each analytical method? Are some of these techniques specifically used to obtain certain characteristics of the CRs? I think you can organize better this section by making clear what is the benefit or output that you are looking for from each method. For example, if some techniques provide the elemental composition, you can group them together. If other methods give you the size and optical features, group them together, and so on. You can even create a table to summarize the different methods applied and the results that you get from each of them.

4. RESULTS AND DISCUSSION

Having a table that summarizes all the results would be of great value. Though you present some figures, they are general and just address some examples of your results. You need a way to put together all your results in a chart or graph. I think that you are missing a huge opportunity on reporting the quantities that were obtained from the analytical methods followed by a comprehensive interpretation of the results. That will make the paper more complete and robust.

What about a plot that shows the relationship between the time (years) of the CRs and the type of aggregates formed/concentration of PHEs? That would be an interesting analysis.

Lines 192-195. These sentences are very hard to follow. I suggest rewording these lines. I don't see the goal of these sentences. Are you trying to back up an argument about the CRs impact?

4.1 Major Minerals and NPs

Do any of the analytical techniques that you applied offer the abundance of these minerals in the samples? I was very interested in seeing the percentages of the different mineral phases in the samples. For example, How did you determine that quartz was the most abundant mineral? (Line 233) I think you have the data and you should report it.

Line 236-243. The facts that you present here are true. However, assuming that the UFPs are respirable (i.e. particle size <4 μm). They need to be aerosolized and then inhaled. If the UFPs in the CRs are stable and deposited, they don't present any hazard for people. Here, you should explain the mechanisms that might put the UFPs suspended in the air available for people to breathe. Otherwise, your argument falls short on claiming that quartz UFPs are a potential health hazard.

CONCLUSIONS

Consider addressing some of the following questions when setting up your conclusions:

What are the key findings of the study? What can you say about the composition/characteristics of the Colombian CRs? Are they hazardous? To which degree? What specific impacts generate the CRs? Can you quantify them? What future work do you recommend? Did you identify areas that contained more toxic/hazardous CRs? What insights can you get from your data? How do you support the points that you made with the data obtained? Did you find differences in CRs composition compared to other studies?

Author Response

Reviewer’s #4

Comments and Suggestions for Authors

OVERALL ASSESSMENT

The paper addresses an important topic that is currently very relevant and worth publication. However, some aspects need to be addressed/complemented.

In terms of the English language, some parts of the text are hard to follow and need editing.

Overall, I think that the paper is lacking more details and data analysis in the results and discussion section. The paper offers a very detailed qualitative description of the composition, characteristics, end effects of CRs, but falls short providing quantitative results, which are key to drawing conclusions and explaining the social-environmental impacts objectively.

In the following sections, I will provide a more detailed explanation of the opportunities for improvement.

ABSTRACT

The structure of the abstract can be improved. Instead of starting with the methods in the first two sentences, start with the background/problem statement (third sentence). Additional background information will make the abstract stronger. This is an example of a better structure for your abstract:

With the possible increase in mining activities and recently projected population growth in Colombia, large quantities of nanoparticles (NPs) and potentially hazardous elements (PHEs) will be of major concern (why is it a major concern? You can be a little more specific here) to indigenous residents (only indigenous residents? What about nearby communities?)...

After this opening, I recommend adding a sentence that includes a little more background (e.g., importance/benefits of gaining knowledge of CRs characteristics or constituents).

...This study highlights the need to regulate the pollution from Colombian mining activities that comply with regional regulations and global strategies. Colombian coals rejects (CRs) from the Cesar Basin Colombia, were studied primarily by advanced electron microscopic and analytical procedures. Therefore, the goal of this research is to evaluate the role of NPs in the alteration of CRs structure in a renewed zone at Cerrejon coal area (La Guajira, Colombia) through advanced electron microscopic (AEMs) methods. The objective of the analysis is to evaluate the incidence mode of nanoparticles (NPs), which contain potentially hazardous elements (PHEs)...

From here the structure of the abstract looks good.

Note: Remember to always write out all the first in-text references to an acronym, followed by the acronym itself enclosed by parentheses. 

Author’s Response: The abstract section has been reworked and improved on significantly.

  1. INTRODUCTION

Line 33. I believe the first sentence is a serious statement that can be misinterpreted and considered subjective. To avoid that, I suggest using facts and data to back up this point. Instead of saying "Coal mining is one of the worst human and environmental tragedies on the planet" you can say something like: "Coal mining activities have caused several environmental and social impacts such as..." and highlight some examples of disasters, or tragedies that unfortunately are not rare in this industry.

Author’s Response: We have recast the sentence in the introduction section.

Lines 45-57. What is the goal of these sentences? Why are you giving me this information? How does it relate to the understanding of CRs? You might want to add a sentence to elaborate on this.

Author’s Response The authors have elaborate on the sentences on lines 45-57.

I wanted to see what has been done in this field. Has anyone tried to study the coal rejects to identify potential hazardous elements before? Are the results from those studies consistent with your results? If there are no such studies, I think this is a great opportunity to elaborate on the importance of the study and your contributions to the field. I think that you kind of address this in line 53 but you should objectively say that no work has been done in this area.

Author’s Response: We have justify the need for the study and provide sufficient background and previous work on the study.

Good job stating the objectives of the study very clearly at the end of the introduction.

Line 51. Here you introduce CR aggregates but what is their importance? Why are you studying the CR aggregates? Do they tell any valuable information about CRs? In the following sections of the paper you address the aggregates again but it is is unclear why you are studying them. Elaborate more on this. 

Author’s Response: We have provided sentences on the importance of coal mine rejects aggregate.

  1. STUDY AREA

Line 97. Did you mean soil or earth? "The earth is categorised as Entisol, which showed..."

Author’s Response: The authors have changed earth to soil.

  1. MATERIALS AND METHODS

3.1 Sampling Strategy

It would be great to have a figure here. It will be helpful to fully understand how the samples were collected and what methods were used to that end. I recommend using a flow chart ideally with images or illustrations showing the sampling procedure step by step. That will make the methods more reproducible as well. How does the Teflon circle look like? How big it is? It is hard to understand the 1 cm intervals, depths, and 2 cm intermissions without a graphic that explicitly shows it. If no figures/graphics are available, elaborate more to make sure your procedures are very clear and easy to follow.

Author’s Response: Authors have provided the purpose of using Teflon circle in the sampling section.

Line 146. What do you mean by 3, 9, 15, and 21 years? How did you determine these years? Is this the time that the CRs have been sitting in that specific zones? You want to be very clear so there is no room for misinterpretation.

Author’s Response: 3, 9, 15, and 21 years are ages of coal reject disposal dumps.

Line 149-151. The CRs from coal seams are not clear for me. Did you take material directly from the coal seams at the mine site? Did you take samples from material that was rejected but not disposed yet? What criterion did you follow to classify the material from the coal seam as CRs? Elaborate more to clear any possibility of doubt here.

Author’s Response: The coal rejects were obtained from different disposal dump with various ages.

Line 152. The word "showed" seems odd here. "Quantities were showed separated? on elected portions and powdered..."

Author’s Response: The word “showed” has been changed to separated.

3.2 Analytical Methods

You used an incredibly large number of methods here which is great, but it looks very confusing and disorganized. Why are you applying these different methods in the first place? Do you want to compare the results of each analytical method? Are some of these techniques specifically used to obtain certain characteristics of the CRs? I think you can organize better this section by making clear what is the benefit or output that you are looking for from each method. For example, if some techniques provide the elemental composition, you can group them together. If other methods give you the size and optical features, group them together, and so on. You can even create a table to summarize the different methods applied and the results that you get from each of them.

Author’s Response: The analytical methods section has been rearranged and reordered for better clarity.

  1. RESULTS AND DISCUSSION

Having a table that summarizes all the results would be of great value. Though you present some figures, they are general and just address some examples of your results. You need a way to put together all your results in a chart or graph. I think that you are missing a huge opportunity on reporting the quantities that were obtained from the analytical methods followed by a comprehensive interpretation of the results. That will make the paper more complete and robust.

Author’s Response: We have separated the results from discussion. This make the manuscript easy to follows and prevent suppression of the significant findings.

What about a plot that shows the relationship between the time (years) of the CRs and the type of aggregates formed/concentration of PHEs? That would be an interesting analysis.

Author’s Response: Thank you for this suggestion. The study is a work in progress thus other aspect of the study will take care of your suggestion.

Lines 192-195. These sentences are very hard to follow. I suggest rewording these lines. I don't see the goal of these sentences. Are you trying to back up an argument about the CRs impact?

Author’s Response The sentence has been reworked.

4.1 Major Minerals and NPs

Do any of the analytical techniques that you applied offer the abundance of these minerals in the samples? I was very interested in seeing the percentages of the different mineral phases in the samples. For example, How did you determine that quartz was the most abundant mineral? (Line 233) I think you have the data and you should report it.

Author’s Response Siroquant analysis technique was used in this study to determine the proportions of mineral phase identified.

Line 236-243. The facts that you present here are true. However, assuming that the UFPs are respirable (i.e. particle size <4 μm). They need to be aerosolized and then inhaled. If the UFPs in the CRs are stable and deposited, they don't present any hazard for people. Here, you should explain the mechanisms that might put the UFPs suspended in the air available for people to breathe. Otherwise, your argument falls short on claiming that quartz UFPs are a potential health hazard.

Author’s Response The ultrafine particle of quartz depending on the size and shape can be easily aerosolized and inhaled.theby significantly affects the human health.

CONCLUSIONS

Consider addressing some of the following questions when setting up your conclusions:

What are the key findings of the study? What can you say about the composition/characteristics of the Colombian CRs? Are they hazardous? To which degree? What specific impacts generate the CRs? Can you quantify them? What future work do you recommend? Did you identify areas that contained more toxic/hazardous CRs? What insights can you get from your data? How do you support the points that you made with the data obtained? Did you find differences in CRs composition compared to other studies?

Author’s Response: We have reworked the conclusion section according to the suggestion of the reviewers.

Round 2

Reviewer 1 Report

The theme addressed by the authors is of great environmental and social importance and is fully part of the topics covered by Sustainability. Compared to the first version, the text has been significantly improved and the various problems highlighted above have been addressed, in particular as regards sampling, analytical methods and the distinction between results and discussion. In its current form, the paper looks mature for publication - English is good, I only noticed a few typos.

Reviewer 3 Report

Dear Authors,

since you have corrected all the comments I made, I see no reason not to recommend your manuscript for publication.